# Extending Robust Adversarial Reinforcement Learning Considering Adaptation and Diversity

**Hiroaki Shioya, Yusuke Iwasawa, Yutaka Matsuo**
The University of Tokyo
Bunkyo-Ku,Tokyo,Japan
{shioya,iwasawa,matsuo}@weblab.t.u-tokyo.ac.jp

## Abstract

We propose two extensions to Robust Adversarial Reinforcement Learning. (Pinto et al., 2017) One is to add a penalty that brings the training domain closer to the test domain to the objective function of the adversarial agent. The other method trains multiple adversarial agents for one protagonist. We conducted experiments with the physical simulator benchmark task. The results show that our method improves performance in the test domain compared to the baseline.

## 1 Introduction

Deep reinforcement learning has developed in various domains such as video games (Mnih et al., 2015), Go (Silver et al., 2016), and robots (Chebotar et al., 2017). However, deep reinforcement learning requires so many trials that it is hard to apply to the real world problems.

To avoid high sample-complexity problems, transfer from simulation to the real world may be a promising solution. However, the physical properties and physics in a simulator are not the same as the real world. This modeling error often makes the policy fail to generalize to the test domain.

To mitigate this problem, we can train an agent in an environment as close as possible to the test domain (Hanna, 2017), train in diverse domains as possible(Yu et al., 2017)(Peng et al., 2017), or in a difficult domain.(Pinto et al., 2017)(Rajeswaran et al., 2016) Robust adversarial reinforcement learning (RARL) (Pinto et al., 2017)is training jointly protagonist agent and adversarial agent to learn a robust policy against adversarial disturbance. However, this approach does not consider how close the training domain is to the test domain or the diversity of the domains.

We propose two extensions to RARL that take these two factors into account. One is to add a penalty on the difference between the transition function of the training domain and that of the test domain to the objective function of the adversarial agent. This penalty term forces the training domain not only to be hard for the protagonist, but also to be close to test domain. The other method is to train multiple adversarial agents simultaneously for one protagonist. Multiple adversarial agents are capable of giving the protagonist more diverge samples than a single adversary.

We conducted experiments with the MuJoCo benchmark task in OpenAI gym. Our experiment shows that our method has improved performance in the test domain compared to the baseline.

## 2 Method

### 2.1 Add Penalty for Difference from Test Domain

We propose a method that force the adversarial agent to sample not only hard domains but also close ones to the test domain. In original RARL, the adversarial agent minimize reward: $\min_{\pi_{adv}} R$ where R is sum of the reward $r_t$ of the episode and $\pi_{adv}$ is a policy of the adversarial agent. We add this objective to a penalty term on the difference of the transition function of the training domain which is changed by adversarial agent and that of the test domain, so as to train the protagonist in

the area close to the test domain. As a result, the adversarial agent optimize the following equation. $\min_{\pi_{adv}} R + \lambda L(S,T)$ where $L(S,T)$ is a penalty term and $\lambda$ is a hyper parameter.

The penalty term $L(S,T)$ is calculated as follows: $L(S,T) = \frac{1}{N}\sum_i^N \|s_{t+1} - T_t(s_t,a_t)\|^2$ where$(s_t, a_t, s_{t+1})$ is state-action-next state tuples from the training domain and $T_t$ is a transition function in the test domain. $T_t$ is learned by using trajectories sampled by running the current policy in the test domain.

The pseudocode of the proposed method is Algorithm1.

---
**Algorithm 1** proposed algorithm

---
Initialize policy parameter $\theta_0^{pro}$ for $\pi_{pro}$ and $\theta_0^{adv}$ for $\pi_{adv}$
**for** $i = 1, 2...N_{itr}$ **do**
  $\theta_i^{pro} \Leftarrow \theta_{i-1}^{pro}$
  **for** $j = 1, 2...N_{pro}$ **do**
    sample $N_{traj}$ trajectory $(s_t^i, a_t^{ipro}, r_t^{ipro})_i, i = 1...N_{traj}$ from training domain
    $\theta_i^{pro} \Leftarrow$ PolicyOptimizer$((s_t^i a_t^{ipro} r_t^{ipro})_i i = 1...N_{traj}, \theta_i^{pro})$
  **end for**
  $\theta_i^{adv} \Leftarrow \theta_{i-1}^{adv}$
  **if** $mod(N_{itr}, K) = 0$ **then**
    sample $N_{traj}$ trajectory $(s_t^i, a_t^{ipro}, r_t^{ipro})_i, i = 1...N_{traj}$ from test domain
    refit transition function for test domain
  **end if**
  **for** $j = 1, 2...N_{adv}$ **do**
    sample $N_{traj}$ trajectory $(s_t^i, a_t^{ipro}, r_t^{ipro}, a_t^{iadv}, r_t^{iadv})_i, i = 1...N_{traj}$ from training domain
    $\theta_i^{pro} \Leftarrow$ PolicyOptimizer$((s_t^i, a_t^{ipro}, r_t^{ipro}, a_t^{iadv}, r_t^{iadv})_i, i = 1...N_{traj}, \theta_i^{pro})$
  **end for**
**end for**

---

## 2.2 ROBUST ADVERSARIAL REINFORCEMENT LEARNING WITH MULTIPLE ADVERSARY

We propose a method that trains multiple adversarial agents for one protagonist. Multiple adversarial agents are capable of giving the protagonist more diverge samples than single adversary.

To make the policy more robust, the simplest way is to use harder samples from all adversarial agents. However, in our experiment, this method stops the learning process. To address this problem, we prioritize the sample on which learning is progressing. In order to judge the progress of learning, we use linear regression by using samples from each adversarial agent of the most recent T iterations and use its regression coefficient.

Considering both how hard the sample is and the progress of learning, we rank samples: samples whose regression coefficient is positive higher than samples whose regression coefficient is negative. the internal ranks of the positive sample and negative sample were higher as the reward is higher.

According to the rank, We choose samples stochastically, which also prevents from using only too hard samples and makes samples more diverge. The probability of selection for each sample is given by the following equation.$p_j = \lambda_j^\alpha / \sum_k \lambda_k^\alpha$ where$\lambda = 1/rank$, and $\alpha$ is a hyper parameter that adjusts the importance of the ranking.

Also, to make the policies of the multiple adversarial agents more diverge, we add a penalty of the objective. the objective function of the adversarial agent is as follows: $\min_{\pi_{adv_i}} R - \gamma \sum_{ij} KL(\pi_{adv_i}(a_t|s_t)\pi_{adv_j}(a_t|s_t))$ where $KL(\pi_{adv_i}(a_t|s_t), \pi_{adv_j}(a_t|s_t))$ is the KL divergence of the i th adversarial agent and the j th adversarial agent policy and $\gamma$ is a hyper parameter that adjusts the weight of the penalty term.

# 3 EXPERIMENT

## 3.1 EFFECTIVENESS OF PENALTY FOR DIFFERENCE FROM TEST DOMAIN

We conducted experiments in the physical simulator MuJoCo task provided in OpenAI gym(Brockman et al., 2016). We train policies in the environment of the default parameter of the

simulator, and test the policy in one different test domain with which our proposed method (and Adaptation, mentioned later) interacts. We use Trust Region Policy Optimization(TRPO)(Schulman et al., 2015) as the policy optimizer.

We compared the proposed method with the following methods. RARL(Pinto et al., 2017), Adaptation: We use the objective function of the adversarial agent represented in Section 2.1, omitting the first term for minimizing the reward, TRPO-target: TRPO trained only in the target task with the same number of iteration as the proposed method used to collect samples of the test domain.($= N_{itr}/K$) We use $N_{itr} = 500$, $K = 10$.

|  | Hopper | | | Walker2D | | HalfCheetah | |
|---|---|---|---|---|---|---|---|
|  | test domain 1 | test domain 2 | test domain 3 | test domain 1 | test domain 2 | test domain 1 | test domain 2 |
| RARL | 1349.6±471.±0 | 1544.8±679.1 | 1555.9±694.5 | 842.2±577.4 | 461.9±354.2 | 1608.1±748.6 | **1608.9±755.1** |
| Adaptation | 1541±351.6 | **2097±295.2** | 1049.6±91.4 | 453.8±266.4 | 596.5±290.8 | **1991.1±718.1** | 1553.7±373.8 |
| TRPO-target | 902.3±681.3 | 811.5±606.6 | 806.6±608.4 | 237.3±70.7 | 296.1±61.5 | 90.6±60.7 | 40.9±116.9 |
| proposed method | **2077.5±462.1** | 2022.1±673.2 | **1632.1±624.6** | **1410.8±450.5** | **708.1±326.7** | 1101.0±70.7 | 1250.7±360.7 |

Table 1: the average reward ± standard deviation of each method on each task and test domain.

The results are presented in Table 1. Our method outperforms RARL in 5 out of 7 tasks and all the other method in 4 out of 7 tasks.

## 3.2 EFFECTIVENESS OF MULTIPLE ADVERSARY AND SAMPLING METHOD

We conducted experiments in the MuJoCo benchmark task. We train policies in the environment of the default parameter of the simulator, and evaluate the policy in 169 different test domains.

We compared the following methods.RARL(Pinto et al., 2017), max: Choose samples with lower rewards in descending order, mean: Equally select samples from all adversarial agents. This corresponds to $\alpha = 0$ in the proposed method, soft: the proposed method. We set $\alpha = 1$.

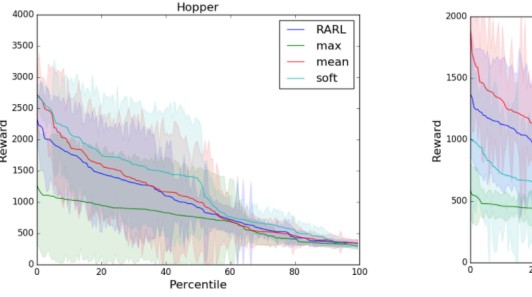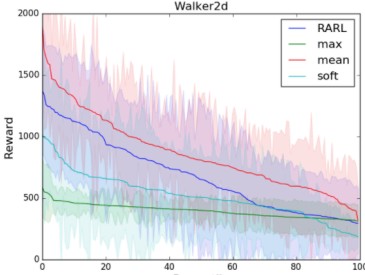

Figure 1: Percentile plot of different test domains. We run learned policies in 169 different test domains and plot the results in descending order of rewards.

Experimental results are shown in Figure 1. It shows that mean has higher generalization performance than RARL. This indicates the effectiveness of using multiple adversarial agents. Soft outperforms mean in one task and not in the other. This suggests that the soft weighting can adjust the performance of this method. Max get lower rewards in almost all the tasks than other methods not only in easy environments but also in difficult ones. this suggests that when we use the max method, protagonist fail to progress learning rather than learn too robust (or conservative) policy.

## 4 CONCLUSION

In this paper, We propose two extensions to RARL: One is to add a penalty that brings the training domain closer to the test domain to the objective function of the adversarial agent, the other method trains multiple adversarial agents for one protagonist. Our experiment shows that our method improves performance in MuJoCo physical simulator task compared to the baseline.

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
