# OpenReview forum: "Extending Robust Adversarial Reinforcement Learning Considering Adaptation and Diversity"
_ICLR.cc/2018/Workshop — Accept_

### Official Review · AnonReviewer1 · 2018-03-08

**Rating:** 6
**Confidence:** 3

**Review:**

This paper proposes two extensions on the robust adversarial reinforcement learning algorithm by Pinto et al., (1) for encouraging the adversary to move closer to the test-domain, and (2) using multiple adversaries.

Pros:
- interesting approach to extending RARL motivated by sim-to-real transfer.
- promising performance of one of the extensions (penalty for difference from test domain)

Cons:
- the paper has lots of typos and grammatical errors.
- there are two disjoint improvements that are presented and evaluated separately, rather than a unified algorithm.
- the paper is motivated from the perspective of sim-to-real, but lacks crucial experimental details for determining if it would be useful for this setting. [e.g. how much test-domain data is used?]
- the results for the second extension (multiple adversaries) does not show a meaningful improvement.

Here are some questions, comments, and suggestions for improvements:
1. For the experiment in 3.1, how much data from the test domain was used? Qualitatively, how much did the training and testing domain differ? Both of these are crucial for understanding if the evaluation is representative of the challenge of transferring from simulation to the real world.
2. The comparison to RARL for the first task doesn't seem particularly fair, since the RARL algorithm wasn't designed for this. Is there another method that would provide a more meaningful comparison?
3. Learning a model T_t would likely be more difficult in tasks with high-dimensional observation spaces. Is there anyway to mitigate this?
4. In Figure 1, what do the shaded regions represent? It seems like the proposed algorithm is only marginally (and insignificantly) better than RARL for the hopper and is worse for the walker.
5. The idea of using multiple adversaries has been previously proposed, e.g. by [1].
6. One of the most common grammatical mistakes was to remove the space before and after parentheses.

[1] https://arxiv.org/abs/1611.01673

---

### Official Review · AnonReviewer3 · 2018-03-09
**A good exploratory work on improving RARL for test performance, but lacks more thorough empirical comparisons**

**Rating:** 5
**Confidence:** 4

**Review:**

The paper proposes two modifications to robust adversarial reinforcement learning (RARL): (1) add penalty that penalizes deviation from test domain dynamics, (2) multiple adversarial agents.

A simplest way to implement (2) is to use an ensemble of adversaries. “mean” plots (alpha=0) in Figure 1 roughly correspond to this (if ignore KL penalties), and exhibit most stable performance improvements over the baseline RARL. The “soft” method (alpha=1) does not achieve consistent improvements. The improvement of “mean” over the baseline is not significant, given that it requires more memory and computation. It’s helpful to include comparison without KL penalties, which is the closest to direct application of ensemble to RARL adversary.

The motivation of (1) is sensible. However, this brings additional assumptions that RARL does not include: collecting samples on test domains during training. Those samples are used for fitting the dynamics model in this paper; however, they can be used in other ways as well: directly use them for protagonist policy update along with RARL learning; use the learned model for additional model-based training. “TRPO-target” in Table 1 is only a limited case of these, and to fully justify the advantage of the proposed approach, alternatives are ideally also compared. The surprisingly good results of “Adaptation” are interesting and may be expanded in results discussion, since this method is quite distinct from RARL: it’s effectively trying to learn interventions to training domains that make the dynamics similar to test domains (based on multi-step errors rather than single-step), and thereby training can generalize. [1] can also be seen as a recent example of this, but depending on how interventions are defined, there are novel directions to pursue.

Overall, it’s slightly weak for acceptance. The main criticism is that given both proposals involve additional requirements (more computation & access to test domains), it’s helpful to include the simplest variants of RARL that involve those as base comparisons (naive ensemble & include both RARL + test policy gradient during training).

Given that this training procedure interacts with test domains during learning, it is also possible to frame in few shot learning setups, as in MAML [Finn et. al., 2017], and derive better extensions for RARL there.

Minor points:
- Writing needs more attention (citation brackets & spacing, use of generic terms, e.g. “a difficult domain” -> “adversarial domains”, Sec 3.2. second paragraph spacings...).

Pros:
- clear descriptions of proposed modifications

Cons:
- proposed modifications are relatively straightforward and the improvements are not substantial given additional requirements (more computation; access to test domains).

[1] Konstantinos et. al., 2017. “Using Simulation and Domain Adaptation to Improve Efficiency of Deep Robotic Grasping”.

---

### Official Review · AnonReviewer2 · 2018-03-12
**Compelling extensions to RARL to regularize the adversary**

**Rating:** 7
**Confidence:** 4

**Review:**

The authors' propose a modification of RARL to learn more robust policies by adding two regularizers. The first, to ensure the adversary doesn't deviate to far from expected behavior the and the second is to train multiple diverse adversaries to randomly choose from.  The overall enhancements are intriguing and the existing experiments validate the approach.

It appears the thrust of the first approach is to provide a signal to regularize the amount the adversarial action is allowed to shift the next state.  Using a trained transition model has a couple of interesting side-effects that should be explored 1) when uncertain/beginning training, there will be large error and uncertainty in prediction which will penalize adversarial actions and 2) when in a region that the transition function has converged the penalty enforces small perturbations.

The second regularizer is geared towards training a diverse set of adversaries and then sampling them weighted by how hard they are. Having numerous advaries that are forced to be diverse may help avoid collapse of learning. The details of this approach were unclear, in particular how the regression was performed, so the author's should clarify this approach.

As a meta comment, the two regularization methods appear unrelated and may warrant being discussed in two different publications.

General comments:
Please mention how \lambda affects the performance of the method and what parameter worked best.
There may be a notational issue as it appears that Algorithm 1 doesn't update/train \theta^{adv}.
RARL uses the adversary and protagonist during rollouts.  You might want to clarify that this is the case in Algorithm 1.
Please describe your experiment for the 169 test domains in greater detail including the parameters you are changing and the ranges you are sampling.
Please provide details for reproducibility

---

### Decision · Program_Chairs · 2018-03-20
**ICLR 2018 Workshop Acceptance Decision**

**Decision:**

Accept

**Comment:**

Congratulations, your paper was accepted to the ICLR workshop.